# *Opuntia Ficus-Indica* L. Miller (Palma Forrageira) as an Alternative Source of Cellulose for Production of Pharmaceutical Dosage Forms and Biomaterials: Extraction and Characterization

**DOI:** 10.3390/polym11071124

**Published:** 2019-07-02

**Authors:** Amaro César Lima de Assis, Larissa Pereira Alves, João Paulo Tavares Malheiro, Alana Rafaela Albuquerque Barros, Edvânia Emannuelle Pinheiro-Santos, Eduardo Pereira de Azevedo, Harley da Silva Alves, João Augusto Oshiro-Junior, Bolívar Ponciano Goulart de Lima Damasceno

**Affiliations:** 1Graduate Program of Pharmaceutical Sciences, State University of Paraíba, Campina Grande 58429-500—PB, Brazil; 2Laboratory of Development and Characterization of Pharmaceutical Products, Department of Pharmacy, State University of Paraíba, Campina Grande 58429-500—PB, Brazil; 3Graduate Program of Biotechnology, Laureate International Universities—Universidade Potiguar, Natal 59056-000—RN, Brazil

**Keywords:** celullose, *Opuntia ficus-indica* L. Miller (palma forrageira), pharmaceutical excipients

## Abstract

Cellulose is among the top 5 excipients used in the pharmaceutical industry. It has been considered one of the main diluents used in conventional and modern dosage forms. Therefore, different raw materials of plant origin have been evaluated as potential alternative sources of cellulose. In this context, *Opuntia ficus-indica* L. Miller (palma forrageira), a plant of the cactus family that has physiological mechanisms that provide greater productivity with reduced water requirements, is an interesting and unexplored alternative for extracting cellulose. By using this source, we aim to decrease the extraction stages and increase the yields, which might result in a decreased cost for the industry and consequently for the consumer. The aim of this work was to investigate the use of *Opuntia ficus-indica* L. Miller as a new source for cellulose extraction, therefore providing an efficient, straight forward and low-cost method of cellulose II production. The extraction method is based on the oxidation of the lignins. The obtained cellulose was identified and characterized by spectroscopic methods (FTIR and NMR), X-ray diffraction, thermal analysis (TGA-DTG and DSC) and scanning electron microscopy. The results confirmed the identity of cellulose and its fibrous nature, which are promising characteristics for its use in the industry and a reasonable substrate for chemical modifications for the synthesis of cellulose II derivatives with different physicochemical properties that might be used in the production of drug delivery systems and biomaterials.

## 1. Introduction

Cellulose is one of the most abundant renewable polymers in nature and an important component among biomass derivatives [1,2,3]. It consists of a homopolysaccharide of linear chain, formed by repeated units of cellobiose (two molecules of glucose) linked by glycosidic β-(1-4) bonds [3,4,5]. It has a molecular formula of C6H10O5 and is commercially available as a white, fibrous, non-toxic, biodegradable, water-insoluble and relatively hygroscopic powder at 20 °C [6,7,8,9].

The pharmaceutical industry has a great interest in cellulose as it is still one of the main diluents used in solid dosage forms. Furthermore, due to their low toxicity, good stability, high permeation to water, high glass transition temperature (*T*_g_) and a good compatibility with a large number of active agents, hydrodispersible cellulose esters have been used in the production of nanostructured systems such as nanocapsules, nanofibers and micelles [10,11].

In addition, they are used in the preparation of water-soluble films for administration of drugs through buccal, sublingual, ophthalmic and topical routes. Recent reports have demonstrated the potential use of cellulose in the development of scaffolds for tissue engineering, whose application highlights its versatility and importance in the field of biomedicine [10,11,12,13,14,15].

For its use in the industry, high purity wood pulp has been the main source of cellulose [14,15]. However, alternative sources such as agroindustrial residues from biomass, as well as bacteria, tunicate (marine animals), fungi, algae and invertebrates have been previously reported in the literature [16,17]. In addition, fast growing crops that generate fibrous residues in the agribusiness, such as coconut, cotton, soybean, sisal, sugar cane, corn, rice, beans and mango lumps are generally used as plant sources. In fact, cellulose can be obtained from different parts of the plant such as leaves, fruits, stems and rigid structures like the stem and nest [7,18,19,20,21,22].

The source used, along with the process of extraction, isolation and purification are parameters that influence the characteristics of the obtained cellulose and, consequently, its applications [23,24,25]. The chemical reactivity of cellulose’s hydroxyl groups (–OH) allows the synthesis of derivatives with different physical and chemical properties. Some examples are microcrystalline cellulose and cellulose acetate [26].

In this context, *Opuntia ficus-indica* L. Miller (palma forrageira), which is a plant of the cactus family that has physiological mechanisms that provide greater productivity with reduced water requirement [27,28], is an interesting and unexplored alternative for extracting cellulose as it seems to decrease the extraction stages and therefore obtain higher polymer yields in relation to the other sources and methods. This would decrease the costs involved in the production of cellulose, which might result in a reduced cost for the industry as well as for the final consumer.

Due to the ready availability of *Opuntia ficus-indica* L. Miller in the Cerrado region of Brazil and the ever-increasing interest in cellulose and its derivatives, this work aims to extract, identify and characterize cellulose from *Opuntia ficus-indica* L. Miller, which could be used as an alternative for pharmaceutical excipient and manufacture of biomaterials.

## 2. Methods

### 2.1. Plant Material

Samples of *Opuntia ficus-indica* L. Miller, variety IPA-20, were collected during the morning time at the experimental station of the National Semi-Arid Institute (INSA), 7°16’41’’S; 35°57’59’’W, Campina Grande, PB, Brazil. The cladodes (aerial parts of the plant) were washed with water, sectioned and dehydrated in an air circulating greenhouse (TE394/4 MP, TECNAL, São Paulo, Brazil), under controlled temperature at 60 °C for 12 days until constant weight was achieved. The material was crushed using a rotating-knives mill coupled with a sieve of 20 mm mesh size (EDB-5, DeLeo—Willey, Porto Alegre, RS, Brazil). The final powdered samples were obtained in the experimental laboratories of the National Semi-Arid Institute (INSA) and the other processing stages were carried out in the Laboratory of Development and Characterization of Pharmaceutical Products (LDCPF) of the State University of Paraíba.

### 2.2. Cellulose Extraction Process I: Elimination of Soluble Compounds

For the elimination of the soluble chemical compounds, the ethanol/nitric acid method was used. This method is based on the oxidation of lignin by nitric acid. The powdered *Opuntia ficus-indica* L. Miller was hydrolyzed under reflux by 3 successive portions of a nitric acid: ethanol (20:80, *v*:*v*) mixture. The reaction mixture was changed every hour and the resulting material was washed with distilled water until the resulting washing liquid was colorless. The material was then immersed in a 1 mol·L^−1^ NaOH solution for 24 h followed by washing and neutralization with a 10% acetic acid solution. The resulting material was dried in an oven at 105 °C for 3 hours followed by grinding with pestle and mortar.

### 2.3. Cellulose Extraction Process II: Obtention of Holocellulose

Holocellulose is the product that results from the extraction of lignin, which consists of cellulose and hemicelluloses. The process of lignin elimination uses sodium chlorite as the main reagent and is based on the reaction between lignin and ClO_2_/ClO^−^, which are formed after the redox reactions of ClO_2_^−^ in acid medium.

Holocellulose was obtained by weighing 5.0 g of the crude extract of *Opuntia ficus-indica* L. Miller to which 100 mL of distilled water was added. The mixture was kept in a water bath at 75 °C followed by the addition of 0.5 mL of acetic acid and 0.75 g of sodium chloride.

The same procedure was repeated twice, where 0.5 mL of acetic acid and 0.75 g of sodium chlorite was added at each hour. After three hours of reaction, the mixture was cooled to 10 °C, filtered and washed with distilled water at 5 °C until the fibrous residue was whitish colored. Finally, the fibrous residue was dried at 105 °C for 3–6 h and kept in a sealed desiccator until use.

### 2.4. Extraction of Cellulose

During this process, 10.0 g of holocellulose was transferred to a porcelain dish to which 100 mL of KOH solution at 24% (*w*:*v*) was added. The mixture was kept under stirring using a mechanical stirrer for 15 hours at room temperature and then filtered through a glass crucible with a pre-weighed porous plate. The resulting solid residue was washed with two portions of 1% acetic acid and distilled water until the retentate was neutral followed by a final washing with ethanol. The obtained cellulose was dried at room temperature on glass plates kept away from contamination.

### 2.5. Determination of the Yield of the Extraction Process

The yield of the cellulose extraction process was determined from the mass difference between the starting material (powdered sample) and the final product corresponding to purified cellulose, as expressed in Equation (1):(1)$$R\left(\%\right)=\frac{P_{final}}{P_{initial}}\times 100$$
where *R* (%) = percentage yield; Pfinal = final weight of the obtained cellulose; Pinicial = initial weight of the powdered *Opuntia ficus-indica* L. Miller.

### 2.6. Identification and Characterization of Cellulose

#### 2.6.1. Morphological Analysis by Scanning Electron Microscopy (SEM)

The morphology of the samples was analyzed using a scanning electron microscope, current of 40 mA (Quanta 200F, Munich, Germany). The samples were previously arranged in a thin layer of carbon tape and sputter-coated with gold before morphological analysis by field-emission scanning electron microscopy (FE-SEM) (SCD500, LEICA EM, Wetzlar, Germany) with metallization time of 80 s and mean thickness of 10 nm.

#### 2.6.2. Thermoanalytical Analysis: Thermogravimetry (TG) and Differential Scanning Calorimetry (DSC)

The TG curves were obtained using a thermogravimetric module (Q600, TA-Instruments, New Castle, DE, USA). A 4.0 mg of sample was weighed on alumina crucible and heated up to 600 °C under a nitrogen atmosphere with a flow of 50 mL·min^−1^ as purge gas and a heating rate of 10 °C·min^−1^. Prior to the tests, the instrument was calibrated with calcium oxalate monohydrate standard under the same experimental conditions.

The DSC curves were obtained using a differential scanning calorimetric module (DSC Q20, TA Instruments, New Castle, DE, USA) under a nitrogen atmosphere with a flow of 50 mL·min^−1^, heating rate 10 °C·min^−1^ and a maximum temperature of 400 °C. Then, 2.0 mg of sample was weighed and sealed in an aluminum crucible. The instrument was calibrated with indium (melting of 156.6 ± 0.2 °C) and zinc (melting of 419.5 ± 0.3 °C) standards with purity of 99.99%. The heat flux and the enthalpy were adjusted using the ΔH fusion of indium (28.58 ± 0.3 J g^−1^) under the same after mentioned conditions.

#### 2.6.3. X-Ray Diffraction Analysis

The diffractograms were obtained using an X-ray diffractometer (D8 Advance, Bruker, Karlsruhe, Germany) under room temperature (25 °C), CuKα radiation (1.5418 Å), voltage of 40 kV, current of 30 mA, 2θ values ranging from 5 to 50 °C and speed of 0.2 °C·s^−1^.

The crystalline index (CI) was calculated using Equation (2) as proposed by Segal et al. [29]:(2)$$C.I.=\frac{I200-I\mathrm{am}}{I200}\times 100$$
where I200 is the maximum intensity (200) of the lattice diffraction; and Iam is the intensity diffraction of the amorphous peak.

#### 2.6.4. Fourier Transform Infrared (FT-IR) Analysis

Holocellulose and cellulose powders were analyzed as KBr pellets, where FT-IR spectra were obtained (400–4000 cm^−1^) with 64 scans and spectral resolution of 4 cm^−1^ using a FT-IR spectrometer (Vertex-70, Bruker Karlsruhe, Germany).

#### 2.6.5. NMR Spectroscopy

Holocellulose and cellulose were characterized using an Avance-500 NMR spectrometer (Bruker, Bremen, Germany), operating with a frequency of 500 and 125 MHz for ^13^C and ^1^H analysis, respectively. Following this, 30° pulses (12.5 ms for ^1^H and 7.0 ms for ^13^C) were employed and CDCl_3_ was used as solvent.

## 3. Results and Discussion

### 3.1. Cellulose Extraction Process

The process of extracting cellulose from *Opuntia ficus-indica* L. Miller was followed by visual changes in the texture and color of the product obtained at each step of the extraction (Figure 1). Such changes were confirmed by NMR, FTIR, XRD and FE-SEM analyzes.

Figure 1 shows that the method of extraction used in this study resulted in both purification and bleaching processes. The conversion of “a” to “b” occurred after removing the soluble fractions using an ethanol:nitric acid mixture (80:20, *v*:*v*), where the residue was kept in contact with NaOH solution. The step “b” to “c” represents the elimination of acid-insoluble lignins by a reaction using sodium chlorite and acetic acid. Finally, holocellulose and cellulose fractions are separated (“c” to “d”).

The yield of cellulose extraction from *Opuntia ficus-indica* L. Miller was 8.4 ± 0.5%, which is lower than the estimated cellulose content (21%) in this plant. The succulent nature of *Opuntia ficus-indica* L. Miller, with its high extractable content (retentate), high levels of ashes, low dry matter content (11.69 ± 2.5%) and its low amount of fibers are some of factors that may have contributed to the inferior yield in relation to the total estimated amount of cellulose [30,31].

### 3.2. Morphology

The SEM micrographs of holocellulose and cellulose, both obtained from *Opuntia ficus-indica* L. Miller, show fibrous surfaces which is characteristic of lignocellulosic materials (Figure 2). Holocellulose’s morphology is represented by a mixture of irregular, rough and fibrous-looking aggregate particles. On the other hand, cellulose appears as individualized structures with elongated morphology in the form of interconnected rods, forming organized arrangements with smooth roughness along the fiber, whose characteristics are best observed in the higher magnitude micrograph (5000×).

The occurrence of a more defined and individualized fibers in the cellulose extracted from *Opuntia ficus-indica* L. Miller can be attributed to the chemical and structural changes induced after chemical treatment of holocellulose, which are further evidenced by FTIR and XRD. According to Rouhou et al. [32] such modifications in the surface topography may be a result of the extraction and purification processes used to obtain the final product.

### 3.3. Thermoanalytical Analysis

#### 3.3.1. Thermogravimetry

The thermogravimetric curves and their derivatives (TGA-DTG) for cellulose (Figure 3) show its decomposition into three main events. The first event (80–100 °C) is attributed to the loss of water and the volatilization of low molecular weight components such as residual solvents. The second and third events are attributed to the decomposition of cellulose’s polymeric chains, which are respectively represented by two mass losses: 285–345 °C with 64.12% of mass loss and 418–452 °C with mass loss of 7.4%, generating a residue of 17.2%. 

Figure 3 shows that the highest extent of mass loss was observed at 340 °C, which is very similar to the temperature found by Jonoobi [33] for purified cellulosic fibers and by Harini et al. [3] for cellulosic fibers extracted from banana peel. 

Previous report on the TGA-DTG curve for holocellulose [34] that a mass loss occurred around 220 °C due to the thermal decomposition of the polysaccharide. On the other hand, lignocellulosic materials exhibited decomposition steps that start within the 200–260 °C range, which are attributed to the thermal depolymerization of hemicellulose or pectin, whose processes are not observed in purified cellulose [35].

#### 3.3.2. Differential Scanning Calorimetry (DSC) 

The DSC curve for the cellulose extracted from *Opuntia ficus-indica* L. Miller presents two main events (Figure 4) as observed in the TG/DTG curves. Initially, an endotherm event with a peak near 75 °C is observed, which may be related to either the residual moisture or the inter/intramolecular bonded cellulosic chains [19].

The second event is observed in the temperature range of 280–340 °C (Figure 4) with an energy value of 140.6 J·g^−1^, which corroborates the mass loss shown in the TGA-DTG thermogram that may be related to the degradation of cellulose [36]. At temperature values beyond 300 °C, the DSC curve presents a heat flux that seems to be attributed to cellulose decomposition or to a depolymerization process, with the formation of 1,6-anhydroglucose or intramolecular transglycolisation with levoglucan formation [37].

No characteristic events of vitreous transition (*T*_g_) are observed. According to Almeida [38], such events may not be observed with some polymers such as cellulose where the decomposition takes place right before or parallel to the glass transition temperature, which may result in masking of such event.

As previoulsy demonstrated by Miao et al. [39], the thermal profile of holocellulose comprises two exothermic events: the first one occurring within the range of 220–315 °C (degradation of hemicelluloses) and the second around 353 °C (decomposition of the cellulose fraction). These results corroborate with our finding and might indicate the purity of the analyzed cellulose.

### 3.4. X-Ray Diffraction

Figure 5 presents the X-ray diffractograms of holocellulose and cellulose, whose diffraction patterns characterize the structural modifications that occurred from holocellulose to cellulose. The X-ray diffraction patterns are typical of semicrystalline materials.

Holocellulose diffractogram presents peaks at 2θ = 12.5, 15.6, 21.3, 22, 26.5 and 34.4, while cellulose diffractogram shows peaks at 2θ = 12, 20, 21.2, 23.6 and 26. Previous studies have demonstrated that the diffractogram peaks for holocellulose are similar to those for cellulose I as the former is a mixture of cellulose and hemicelluloses [40,41]. In addition, the cellulose diffractogram reveals that it belongs to class II, as demonstrated by Astruc and colaborators [42]. The modification of the cellulose structure might have occurred due to the highly concentrated alkaline treatment (KOH solution at 24% (*w*/*v*)) [43], which causes swelling and consequently the weakening of the molecular bonds, resulting in recrystallization of the cellulose fibers [44].

Besides that, broad and diffuse peaks characteristic of amorphous regions are observed in 2θ = 12.1 and 15.6 for cellulose and hollocelulose, respectively. However, after hemicellulose removal, intense and narrow peaks in 2θ = 26.5 appears which relates to crystalline regions attributed to cellulose. In addition, the diffractograms show high intensity peaks at 2θ = 22 (holocellulose) and 21.4 (cellulose) that characterize semicrystalline regions.

Similar diffraction profiles were observed by Jabli et al. [45], in which the authors assigned the broad peaks around 2θ = 15 to amorphous regions, whereas those of greater intensity, which appeared around 2θ = 22, they attributed to the presence of crystalline regions within the polymer structure. According to Zhang et al. [46], these peaks around 2θ = 22 correspond to the packaging of the polymer chains by Van der Walls forces, which are common to most polymeric materials.

The crystalline index revealed a 64% of crystallinity for holocellulose, which is similar to those found for hollocellulose extracted from flax stem, rose stems and banana peels [42,47,48]. However, crystalline index of cellulose II was 83%, which was better than that obtaneid for cellulose II extracted from sago seed shells, sugarcabe bagasse, alfa fibers and red algae waste [44,49].

### 3.5. Fourier Transform Infrared (FT-IR) Spectroscopic Analysis

The FT-IR spectrum of holocellulose (Figure 6) shows a broad band between 3650 and 3207 cm^−1^, which is attributed to stretching vibration of hydroxyl groups (–OH). Bands of medium intensity were observed at 2920 cm^−1^, which are characteristic of C–H stretching of sp^3^ carbon. In addition, the band at 1634 cm^−1^ results from the deformation of water molecules absorbed by the hydrophilic groups that are present along the holocellulose chain [50]. In addition, bands at 1517 and 1152 cm^−1^ are attributed to C=C vibration of aromatic lignans and to C–O stretch of hemicellulose, respectively [46,51,52].

The FT-IR spectrum of cellulose (Figure 6) shows a gradual decrease in the intensity of the bands related to –OH (3650–3207 cm^−1^), CH (2920 cm^−1^), the absorbed water (1634 cm^−1^) and C=C (1517 cm^−1^), which corroborates the elimination of lignans from holocellulose [50]. In addition, the absence of band in 1737 cm^−1^ in the spectrum of cellulose confirms the disappearance of the carbonyl groups (hemicelluloses) [53], corroborating with the events demonstrated in the XRD after the alkaline treatment.

Finally, the decrease in the intensity of the band at 1152 cm^−1^ corroborates the purity of the cellulose obtained from *Opuntia ficus-indica* L. Miller [54,55].

### 3.6. Nuclear Magnetic Resonance (NMR) Analysis

The characterization through FT-IR and XRD of the cellulose obtained from *Opuntia ficus-indica* L. Miller can be corroborated through ^13^C and ^1^H NMR analysis. The ^1^H NMR spectrum of the obtained cellulose shows a multiplet at 4.1–4.2 ppm (Figure 7A), which suggests the presence of osydic units in the sample. In addition, the ^13^C resonance at 77.44 ppm (Figure 7B) can be assigned to cellulose, which corroborates the identity of this polymer in the analyzed sample [56].

The absence of resonances in the 6.0–8.0 ppm and 100–150 ppm regions in the ^1^H and ^13^C spectrum, respectively, suggests that lignins are not present in the extracted sample. In addition, the absence of resonances due to carbonyls on the ^13^C spectrum corroborates the removal of hemicellulose after the purification step of cellulose extraction [5,35].

## 4. Conclusions

This study showed that *Opuntia ficus-indica* L. Miller can be used as an alternative source of cellulose II, which is an excipient of widespread use in the pharmaceutical industry for the production of conventional and modern pharmaceutical forms as well as biomaterials. Cellulose II has better physical-chemical properties (mechanical and thermal) and stability when compared cellulose I due to antiparallel arrangement of hydrogen bonds. In addition, they can be used for producing bioethanol. The results presented in this study will support further studies aiming to optimize the extraction/purification process of cellulose from *Opuntia ficus-indica* L. Miller, with the purpose of obtaining a higher yield final product, which might help to reduce the costs of cellulose for the pharmaceutical industry and consequently for the consumer.

## Figures and Tables

**Figure 1 polymers-11-01124-f001:**
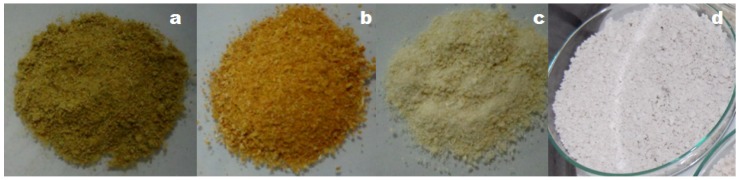
Technological processing of *Opuntia ficus-indica* L. Miller for cellulose extraction. (**a**) powdered plant, (**b**) powdered plant free of impurities, (**c**) holocellulose and (**d**) cellulose.

**Figure 2 polymers-11-01124-f002:**
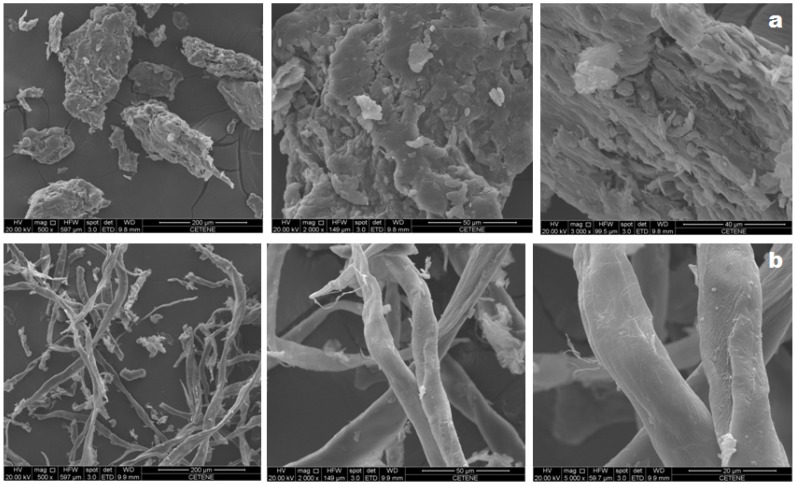
SEM micrographs of holocellulose (**a**) and cellulose (**b**). The magnitudes of the micrographs are 500×, 2000× and 5000×, from left to right.

**Figure 3 polymers-11-01124-f003:**
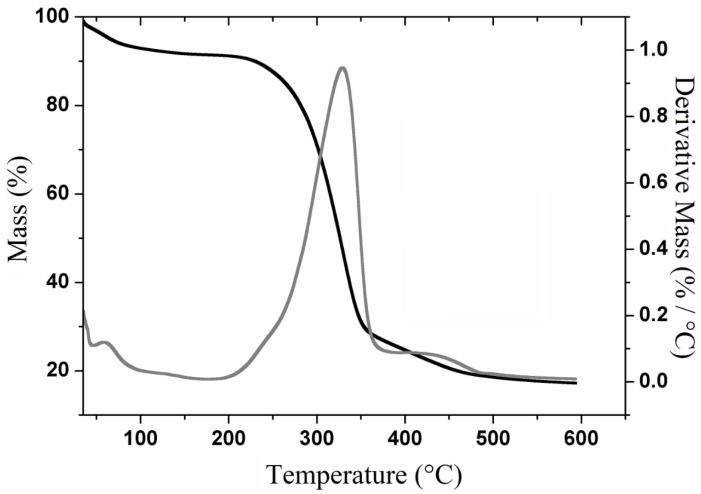
Thermogravimetric curves and their derivatives (TGA-DTG) curves for cellulose extracted from *Opuntia ficus-indica* L. Miller.

**Figure 4 polymers-11-01124-f004:**
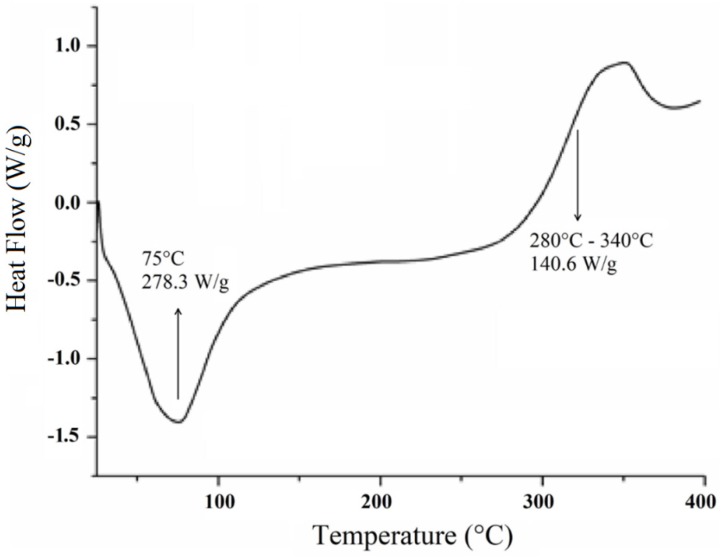
Differential scanning calorimetry (DSC) curve for cellulose extracted from *Opuntia ficus-indica* L. Miller.

**Figure 5 polymers-11-01124-f005:**
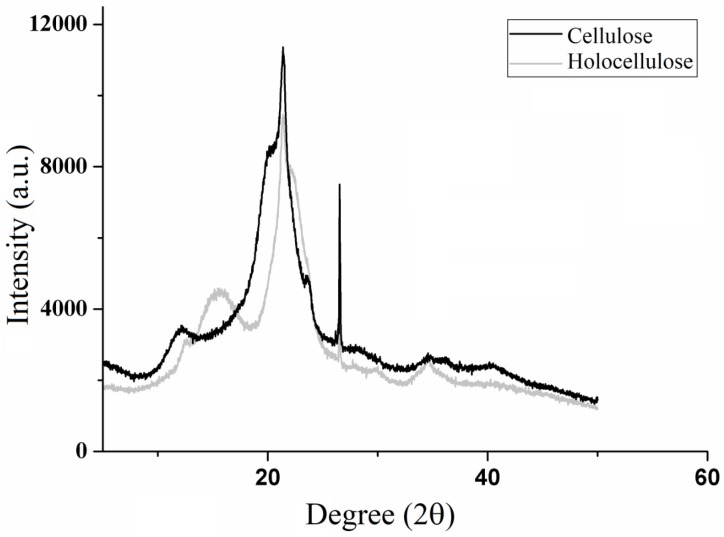
X-ray diffratograms of holocellulose and cellulose.

**Figure 6 polymers-11-01124-f006:**
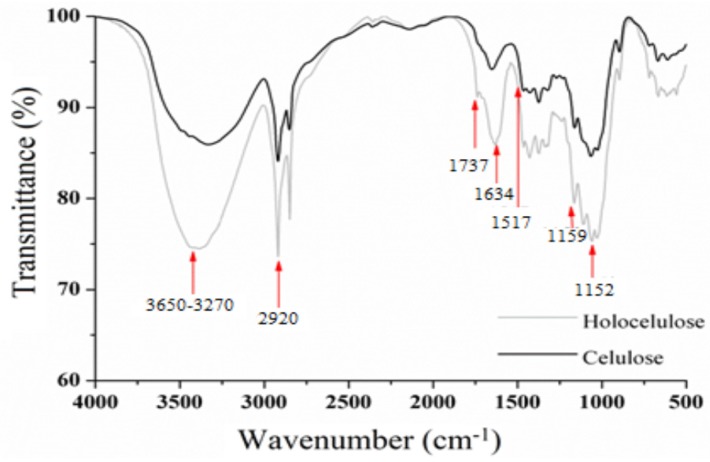
Fourier transform infrared (FT-IR) spectra of holocellulose and cellulose.

**Figure 7 polymers-11-01124-f007:**
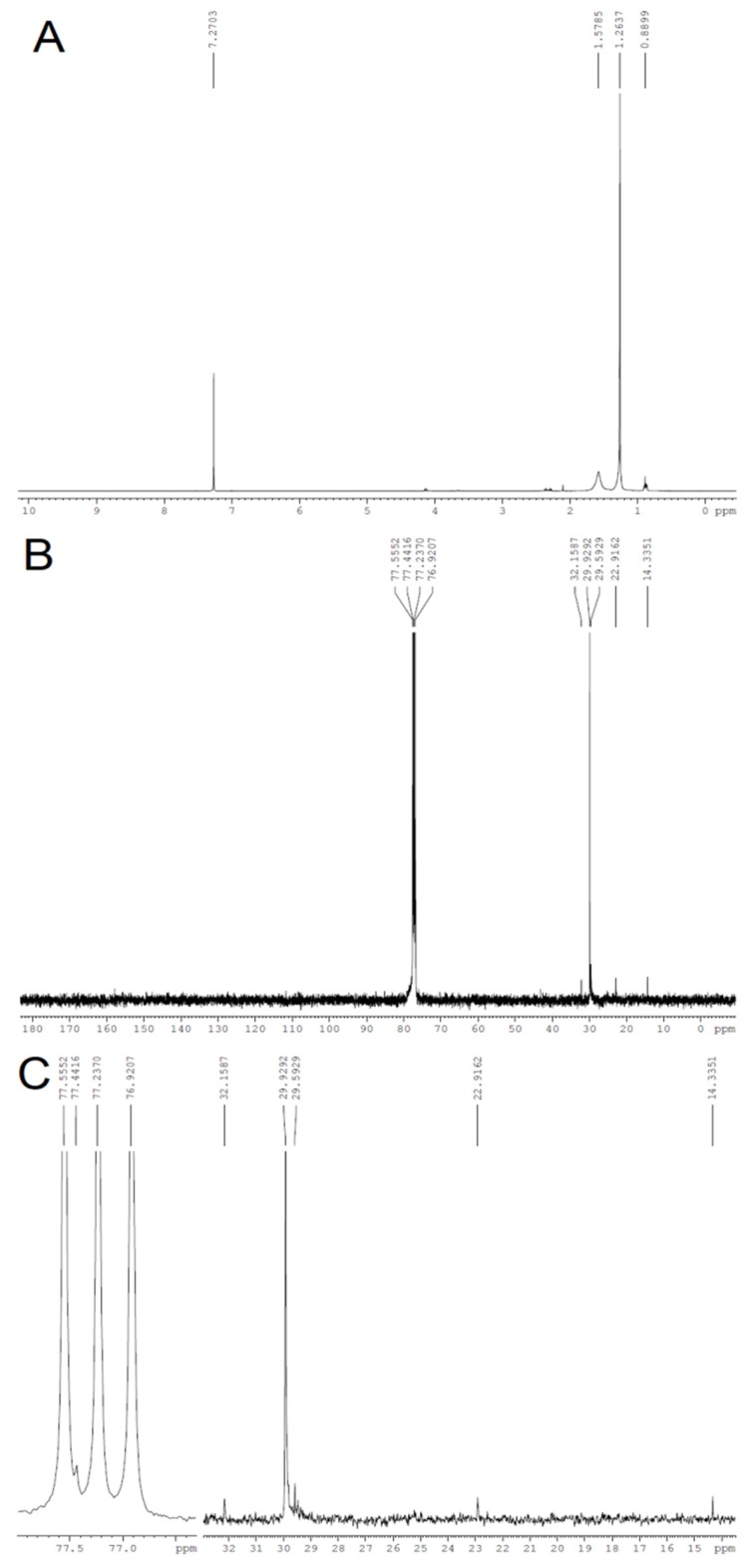
^1^H (**A**), ^13^C (**B**) and amplified ^13^C-NMR (**C**) spectra of cellulose obtained from *Opuntia ficus-indica* L. Miller.

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
