# Peer review of "Opuntia Ficus-Indica L. Miller (Palma Forrageira) as an Alternative Source of Cellulose for Production of Pharmaceutical Dosage Forms and Biomaterials: Extraction and Characterization"

_polymers, 2019, doi:10.3390/polym11071124_

Reviewer 1 Report

All the test should be conducted for the native fiber and to be compared with others. 

In XRD data the hollocellulose peak at 22 degree may be for cellulose II also as reported in different literature. The planes are not mentioned in XRD data.

Transformation of cellulose in different polymorph by treatment should be investigated. Crystallinity of cellulose should be determined by XRD for prediction of applicability of the generated cellulose.  

FTIR peak value should be given in tabulated for and removal of the components from the fiber should be indicated by comparing with the native value. 

In conclusion application is predicted but observed property is not mentioned. 

Author Response

Thanks to the referee comments. We would like to thank the reviewer for  the opportunity of improve our manuscript. Several points in the manuscript was modified to better understanding of this study. Correction of the language was done.  The change is highlighted in the yellow in the manuscript text".

Reviewer 2 Report

The authors used Opuntia ficus-indica L. Miller as a new source for extracting cellulose by oxidation of the lignins. The extracted cellulose was identified by FTIR, NMR, X-ray diffraction, TGA-DTG, DSC and SEM. The manuscript was well organized. However, some points need to be clarified in the manuscript before being recommended to be published in the journal. Therefore, I would like to suggest a minor revision. If the requested points listing bellow could be well addressed and supported in revised version, the publication will be further considered.

1.      Did you use any analytical method to confirm the complete elimination of lignin?

2.      Better to include the DSC and TGA thermograms of holocellulose in the manuscript for comparison.

3.      In FT-IR results, authors mentioned that the absence of bands between 1850-1650 cm−1 244 in the spectrum of cellulose confirms the disappearance of the 245 carbonyl groups in the sample. But there is not much difference in the FT-IR spectra of cellulose and holocellulose. Please give the magnified spectra in the region between 1850 and 1650 cm-1 to confirm the disappearance of peaks between this region.

Author Response

(The authors gave the same response as above.)

Reviewer 3 Report

This work by Goulart de Lima Damasceno, has extracted, identified and characterized cellulose from Opuntia ficus-indica as an alternative for pharmaceutical excipient and manufacture of biomaterials. 

The paper is well written, the experimental set up is adequately realised and the results appear interesting and useful to the readership of the journal.

Author Response

We would like to thank the reviewer for their comments concerning our work

Reviewer 4 Report

The manuscript is very well written and describes the extraction of purified cellulose from a plant of the cactus family and the characterization of the obtained material by SEM, thermal analysis, FTIR and NMR. I have only a few points that must be addressed by the authors.

Line 92. I guess you mean “pestle and mortar”;

Line 110. Maybe “retentate” instead of “filtrate”;

Lines 116 & 117. “initial” instead of “inicial”;

Line 129. Why did you perform TG under air flow while DSC was under nitrogen gas?

Line 153 and some other parts. Opuntia ficus-indica is written in italic up to this line. I guess this is the plant’s species name;

Line 167. “Filtrate” appears again;

Line 168. I wonder why the error is stated with 3 significant figures. Usually it requires one significant figure or two when the leading number is 1 or 2.

Lines 226 & 229. The cited references are not numbered. Jabli et al. is not even listed in the references.

The discussion concerning XRD needs some revision. From my understanding, the cellulose spectrum shows peaks corresponding to cellulose II crystalline structure (specifically the lump around 2theta=12 and the small shoulder just above 2theta=20) while holocellulose spectrum shows cellulose I peaks (the most intense one and the lump at about 2theta=16). This is because the last chemical treatment with KOH at 24% was concentrated enough to cause crystalline structure change. Please elaborate on that.

Author Response

Thanks to the referee comments. We would like to thank the reviewer for  the opportunity of improve our manuscript. Several points in the manuscript was modified to better understanding of this study. Correction of the language was done.  The change is highlighted in the yellow in the manuscript text".

Round  2

Reviewer 4 Report

I have no concerns about the present manuscript.